# Identification of Carotenoids in Hairless Canary Seed and the Effect of Baking on Their Composition in Bread and Muffin Products

**DOI:** 10.3390/molecules27041307

**Published:** 2022-02-15

**Authors:** El-Sayed M. Abdel-Aal, Lili Mats, Iwona Rabalski

**Affiliations:** Guelph Research and Development Centre, Agriculture and Agri-Food Canada, 93 Stone Road West, Guelph, ON N1G 5C9, Canada; lili.mats@agr.gc.ca (L.M.); iwona.rabalski@agr.gc.ca (I.R.)

**Keywords:** hairless canary seed groats, carotenoid identification, LC-MS/MS, baked products

## Abstract

Carotenoids are essential components in the human diet due to their positive functions in ocular and cognitive health. This study investigated composition of carotenoids in hairless canary seed (HCS) as a novel food and the effect of baking on carotenoids in bread and muffin made from HCS, wheat and corn. Three bread formulations made from wheat and HCS blends were evaluated and compared with control wheat bread. In addition, three low-fat muffin recipes prepared from HCS alone or in blends with corn were assessed. The fate of carotenoid compounds in breads and muffins was monitored after dry mixing, dough/batter formation and oven baking. Carotenoids in products were quantified using UPLC and their identification was confirmed based on LC-MS/MS. Hairless canary seed and corn were fairly rich in carotenoids with a total content of 7.6 and 12.9 µg/g, respectively, compared with wheat (1.3 µg/g). Nineteen carotenoid compounds were identified, with all-*trans* lutein being the principal carotenoid in HCS followed by lutein 3-*O*-linoleate, lutein 3-*O*-oleate and lutein di-linoleate. There were significant reductions in carotenoids in muffin and bread products. It appears that batter or dough preparation causes more reductions in carotenoids than oven baking, probably due to enzymatic oxidation and degradation. Muffin-making resulted in lower lutein reductions compared with the bread-making process. The results suggest that muffins made from hairless canary seed alone or in blends with corn could boost the daily intake of lutein and/or zeaxanthin.

## 1. Introduction

Carotenoids are essential components in the human diet due to their positive functions in ocular and cognitive health [1,2,3,4,5]. Lutein and zeaxanthin are xanthophyll carotenoids commonly found at high levels in dark green vegetables, egg yolks, einkorn wheat, and corn [1]. These pigments constitute the yellow spot in the human retina and are referred to as macular pigments [6]. They also accumulate in the brain across the lifespan [5] and have been found to improve cognitive functions [7,8]. In addition, they have been linked with a reduced risk of age-related macular degeneration and cataracts [1]. Epidemiological studies have shown that the status of macular pigment is strongly correlated with lutein intake [9]. Thus, the availability of diets rich in lutein and/or zeaxanthin is crucial for human health due to their physiological and protective attributes, particularly in the eye and brain.

Special grains such as hairless canary seed and high lutein corn could provide promising ingredients for the development of high lutein functional foods. Hairless (glabrous) canary seed (*Phalaris canariensis* L.) was developed through a successful breeding program to eliminate the potential health threats associated with the presence of tiny siliceous hairs on the surface of the hull of the kernel in the hairy (pubescent) varieties [10]. The hairless canary seed (HCS) was approved by Health Canada as a novel food [11] and received GRAS status (GRAS Notice No. GRN 000529) from the US Food and Drug Administration [12]. HCS is a true cereal grain belonging to the grass family (*Poaceae or Gramineae*), similar to wheat, barley, oat, and other cereals. The grain has a unique nutritional profile being relatively rich in protein (22.7%) and oil (7.7%) as a cereal grain, but its starch content is comparable to other cereal grains at 57.2% [13]. In addition, the protein in HCS is free from gluten, which makes it suitable for the gluten-free food market [14]. Several studies have reported that HCS groats hold great potential as a wholegrain food ingredient and/or sustainable source of starch, protein, and oil [13,15,16]. HCS is also considered a health food ingredient being a good source of bioactive peptides, carotenoids, and phenolic compounds [13,17,18]. Currently, little or no data are available on carotenoid composition in HCS and the effect of processing on carotenoids in the end products. This study was aimed at exploring the detailed composition of carotenoids in HCS. In addition, the effect of baking on carotenoids in bread and muffin prototypes made from blends of hairless canary seed, wheat, and corn was investigated to assess their potential in enhancing the consumption of carotenoids.

## 2. Materials and Methods

### 2.1. Chemicals

HPLC or MS-grade methanol, acetonitrile, methyl tert-butyl ether (MtBE), and hexane, sodium hydroxide, and hydrochloric acid were purchased from Fisher Scientific (Mississauga, ON, Canada). Suprapur formic acid (FA) was purchased from VWR (Mississauga, ON, Canada). High purity carotenoid standards, including lutein 00012453 and Zeaxanthin 00026504 were purchased from ChromaDex (Irvine, CA, USA) and β-cryptoxanthin C6368 and β-carotene C9750 from SigmaAldrich (Oakville, ON, Canada). Lutein mono- and di-esters were prepared in the lab as described in our previous publications [19,20]. The nano pure water was obtained from Milli-Q integral water purification system (Millipore Ltd., Etobicoke, ON, Canada).

### 2.2. Grains and Flours

The yellow hairless canary seed was kindly supplied by the Canary seed Development Commission of Saskatchewan and hard red spring wheat by the University of Saskatchewan (Saskatoon, SK, Canada). Yellow corn flour was purchased from the local market in Guelph, ON, Canada. The yellow hairless canary seeds were milled using the UDY cyclone mill (310-014, UDY Corporation, Fort Collins, CO, USA) into wholegrain flour, while the wheat grains were milled on the Brabender Junior mill (Quadrumat Junior, Brabender Instruments Inc., Duisburg, Germany) into wholemeal flour.

### 2.3. Preparation of Bread and Muffin Products

Two baked products were investigated, bread as a fermented food commonly consumed worldwide and muffin as a non-fermented baked food. Three low-fat muffins were prepared from HCS flour alone or in blend with cornflour at ratios of 1:1 and 1:2 (*w*/*w*). The muffin formula contained 120 g HCS flour or flour mix, 36 g sugar, 1.5 g salt, 3 g baking powder, 2.5 g egg white powder, 5 g canola oil, and 85, 80, 80 g water for 100% HCS flour, 1:1 and 1:2 HCS/corn, respectively. First, the dried ingredients were mixed for 1 min, then water and other ingredients were added and mixed for 2 min. A 25 g batter was dispensed into a paper cup, then baked for 16 min at 204 °C. The sourdough method was used for bread making according to the approved AACCI method 10-11.01 [21]. The Lallemand Inc. Florapan L62 culture starter was kindly provided by Lallemand Company (Montreal, QC, Canada). Three bread formulations were made from wheat and HCS flour blends at a ratio of 85/15, 75/25, and 50/50 (*w*/*w*) in addition to 100% wheat flour as a control treatment. The ingredients were 75 g flour, 50 g sour, 2 g salt, 5 g sugar, 3 g corn oil, 0.5 g dry yeast, and 37.5 g water. The dry ingredients and oil were pre-mixed for 2 min, then sour and water were added, and all ingredients were mixed for 3 min. The dough was placed in a proffer at 35 °C for 60 min, then punched, shaped, and panned. The second proofing was carried out for approximately 50 min until the maximum volume was reached. Bread loaves were baked at 204 °C for 21 min. All baking trials were made at least in triplicate, and subsamples were taken at three technological steps, e.g., dry mix, batter or dough, and freshly baked muffins or breads, and changes in carotenoids were monitored during the baking process. The changes in carotenoids during the baking process were calculated as % difference from their corresponding dry mixes using the following equation:% difference = [carotenoid in dry mix − carotenoid in product]/[carotenoid in dry mix] × 100

### 2.4. Analysis of Nutrients and Prodauct Quality

Moisture, ash, and total dietary fiber in ingredients and finished products were determined according to the approved methods of the AACCI Methods 44-15.02, 08-01.01, and 32-21.01, respectively [21]. Protein content was based on the combustion method using a Nitrogen Analyzer (Flash 2000, Thermo Fisher Scientific, Waltham, MA, USA). External and internal qualities of breads were assessed based on the scoring guidelines of the AACCI method 10-12.01 and the volume of bread loaves was measured by the rapeseed displacement method 10-05.01 [21]. The quality of the muffin products was evaluated based on the measurement of the height of the muffin, internal and external quality scores. Three trained technicians were employed to assess the quality of products. The quality score reported is the average internal quality score (crumb structure, texture, and color, flavour and mouthfeel) and external quality score (shape, smoothness, degree of breakage and shred and crust colour).

### 2.5. Quantification and Identification of Carotenoids

Samples of HCS flours and products (0.5 g) were extracted with 5 mL of water-saturated butanol for 30 sec using a Polytron PT 10–30 homogenizer (Kinematica, Switzerland). The butanol extracts were centrifuged at 3030 g for 10 min and the supernatants were filtered through a 0.2-µm GHP syringe filter (Pall Company, Mississauga, ON, Canada) prior to UPLC analysis. Waters Acquity UPLC H Class with eLambda PDA detection and Carotenoid C30 YMC column (10 cm × 4.6 mm × 3 µm) was used. Ten µL of carotenoid extracts were injected onto the column. The mobile phase was composed of methanol (A) and MtBE (B) using a gradient program: at 0 time 95% B, at 7 min 92% B, at 20 min 75% B, at 29 min 60% B, at 30–33 min 0% B and at 34–36 min 95% B. The column temperature was set at 35 °C. Four authentic carotenoids (all-*trans*-lutein, all-trans-zeaxanthin, all-trans-β-cryptoxanthin and all-trans-β-carotene and lab-prepared lutein mono- and di-ester compounds were used for the identification and quantification of carotenoids [20].

Carotenoid compounds were further identified and confirmed based on LC-MS/MS analysis using Thermo Scientific™ Q-Exactive™ Orbitrap mass spectrometer equipped with a Vanquish™ Flex Binary UPLC System (Waltham, MA, USA). For the identification of carotenoid compounds, extracts were subjected to further clean up to remove proteins and oil in order to concentrate carotenoids. A YMC Carotenoid S–3 µm column (150 × 4.6 mm, Chromatographic Specialties Inc., Brockville ON, Canada) was used. Both APCI and ESI methods were used in this study. The mobile phase for APCI method consisted of solvent A (MeOH) and solvent B (MtBE). The mobile phase for ESI mode consisted of solvent A (99.95% MeOH + 0.05% FA) and solvent B (99.95% MtBE + 0.05% FA). The following solvent gradient was used for both APCI and ESI studies: 0–10 min, 5% to 8% B; 10–25 min, 8% to 35% B; 25–28 min, isocratic 35% B; 28–29 min, 35% to 100% B; 29–31 min, isocratic 100% B; 31–32 min, 100% to 5% B; 32–36 min, isocratic 5% B. The column temperature was set at 35 °C. The flow rate was 0.4 mL/ min. The injection volume was 5 or 10 µL. The UV absorption wavelengths were 475 nm (UV1), 439 nm (UV2), and 450 nm (UV3). UPLC chromatograms depicting the separation of the four carotenoid standards and lutein mono- and di-esters are shown in Figure 1A,B. Positive ionization mode was used for both APCI and ESI study. The optimized APCI conditions were as follows: spray voltage, 5 kV; capillary temperature, 350 °C and auxiliary heater temperature, 400 °C. The optimized ESI conditions were as follows: spray voltage, 3.5 kV; capillary temperature, 263 °C and auxiliary heater temperature, 425 °C. Mass spectrometry data was collected using Full-MS/DDMS2 (TopN = 15) method. Data was visualized and analysed using Thermo FreeStyle™ 1.7 software. 

### 2.6. Statistical Analysis

Baking trials and analyses were carried out at least in triplicate, and the data are expressed as means ±SD (standard deviation) on a dry matter basis. One-way ANOVA was used to determine the effect of the baking process on individual carotenoid compounds, and significant differences between means were assessed using Tukey’s method and considered significant at *p* < 0.05. Statistical analyses were performed using Sigma-Plot version 14.5 (Systat Software Inc., San Jose, CA, USA).

## 3. Results and Discussion

### 3.1. Composition and Identification of Carotenoids in Hairless Canary Seed

Hairless canary seed holds a promise as a novel food for the functional food industries a source of non-gluten protein, bioactive peptides, polyphenols and carotenoids [13]. Since carotenoids play significant roles in the health of the eye and brain, the current study provides detailed information on the composition of carotenoids in HCS and their behavior during the baking process. Carotenoids are made up of eight isoprene units having two classes, carotenes (purely unsaturated hydrocarbons) and xanthophyll or oxygenated carotenoids. The main carotenoids found in HCS comprised of lutein and its mono- and di-esters which belong to the oxygenated carotenoids (Table 1). We identified 19 carotenoid compounds based on their mass, UV-vis and structural properties in comparison with authentic carotenoids (Figure 1C, Table 1). Other oxygenated carotenoids in HCS were zeaxanthin, β-cryptoxanthin and their *cis* isomers. Several *cis* configurations were identified at positions C9, C13, C13′ and C15. The presence of *cis* carotenoids in wheat has been previously reported [22,23].

Both APCI+ and ESI+ modes were employed to ascertain the identity of carotenoids. APCI mode is often a preferred method of carotenoid analysis as it provides more predictable in-source fragmentation; however, ESI mode proved to be more sensitive in the analysis of certain less abundant stereoisomers, e.g., lutein mono-esters. Both methods provided similar fragmentation patterns, especially for unbound carotenoids, and are used interchangeably throughout the discussion. The mass characteristics of each carotenoid and the relative abundance percentage are given in Table 1. The MS spectra of all-*trans*-lutein, the dominant carotenoid found in HCS, are shown in APCI+ (Figure 2A) and ESI+ (Figure 2B) ionization modes. In both modes, lutein appears as a radical molecular ion at *m*/*z* 568, along with in-source fragment ions due to the loss of one or two molecules of water to yield protonated ions at *m*/*z* 551 and 533, respectively. The in-source fragmentation pattern was quite similar in both ionization modes with slight differences in their relative abundance percentages. The dominant ion for lutein was the fragment ion [M-H_2_O+H]^+^ at *m*/*z* 551 as indicated by the percent signal intensity (100%).

Despite zeaxanthin having similar molecular weight and structure as lutein, it exhibited a different fragmentation pattern. The protonated molecular ion [M+H]^+^ at *m*/*z* 569 became the most abundant (100 %), and the water loss fragment ion [M-H_2_O+H]^+^ at *m*/*z* 551 is about half as intense. There is a subtle difference between zeaxanthin and lutein in their chemical structure (e.g., zeaxanthin has two β-ionone rings, whereas lutein has a β-ionone ring and ε-ionone ring) which could elucidate differences in the fragmentation pattern between the two molecules under the same conditions. The ε-ionone ring has a C4′-C5′ double bond adjacent to the 3′ hydroxyl group, which could encourage water loss due to allylic stabilization of the resulting cation [22]. The all-*trans*-β-carotene showed a major protonated molecular [M+H]^+^ ion at *m*/*z* 537. Interestingly, zeaxanthin, β-cryptoxanthin and β-carotene showed the preferential formation of protonated molecular ions in APCI, whereas lutein and its esters favoured the formation of a radical molecular ion. In ESI mode, all carotenoids preferably formed non-protonated molecular ion. Ionization of carotenoid monoesters in both APCI and ESI modes produced predominantly radical non-protonated molecular ions and fragments. With our specific experimental conditions, ESI mode provided a more gentle fragmentation resulting in molecular ion often being the most abundant, whereas APCI mode resulted in more abundant in-source fragments (Appendix A). The exact mechanism of molecular ion formation is determined by mobile phase composition and specific mass ionization conditions. Nevertheless, the fragmentation patterns either in APCI+ or ESI+ can assist in the identification of carotenoid molecules [20,24].

Figure 3 shows the mass spectra of lutein mono- and di-esters in the APCI^+^ and ESI^+^ ionization modes. There were obvious differences in the mass spectra between lutein 3′-*O*-linoleate and lutein 3-*O*-linoleate (Figure 3A,B) arising from the structural differences determined by which moiety is attached to the ε-ionone ring, i.e., the acylated fatty acid at the 3′-position in the former molecule while at the 3-position in the latter one. One expects a more or less similar probability of neutral loss of a fatty acid or water molecule from either end of the lutein molecule. However, the preferred fatty acid or water loss occurs from the 3′-position due to allylic stabilization of the resulting cation generated by cleavage in the ε-ionone ring [20]. Then, subsequent losses of water or fatty acid respectively occur at the 3-position in the β-ionone ring. For that reason, the fragmentation patterns of 3′-*O*- and 3-*O*- acylated lutein molecules are different. The differences in MS fragment patterns enable facile discrimination between the 3′*-O-* and 3*-O-*monoester regioisomers of lutein. Based on the fragmentation patterns of the four isomers for lutein 3′-*O*-linoleate and lutein 3-*O*-linoleate, three of them were lutein 3-*O*-linoleate and the fourth one was lutein 3′-*O*-linoleate. The presence of *cis* peaks (λ_cis_) around 330–340 nm along with their elution order was also employed to confirm the identity of two *cis* isomers of lutein 3-*O*-linoleate (Table 1). 

In a previous study, we confirmed the identity of 3′*-O-* and 3*-O-*monoesters of lutein using ^1^H NMR technique [20]. The ^1^H NMR spectrum of pure lutein showed chemical shifts of 3.85 and 4.12 ppm for the C-3 and C-3′ protons on the β- and ε-ionone rings, respectively. In their study, the C-3 and C3′ proton chemical shift values for the 3-*O*-monopalmitate esters are 5.01 and 4.12 ppm and 3.85 and 5.26 ppm, respectively. This indicates that we can clearly confirm the identity of lutein 3′-*O*-and lutein 3-*O*-mono esters based on NMR data. A study has reported the presence of 4 mono-esters (lutein 3′-*O*-linoleate, lutein 3-*O*-linoleate, lutein 3′-*O*-palmitate, lutein 3-*O*-palmitate) and 4 di-esters (lutein di-linoleate, lutein 3′-*O*-linoleate- 3-*O*-palmitate, lutein 3′-*O*-palmitate-3-*O*-linoleate and Lutein di-palmitate) in the endosperm of tritordeum, a novel cereal, using LS-MS APCI positive mode [25]. As seen in Table 1 all-*trans* and *cis* carotenoids exhibited three absorption peaks λ_I_, λ_II_ (λ_max_) and λ_III_, at approximately 412–424, 436–450 and 464–476 nm, plus λ*_cis_* at 330–340 nm only in the case of *cis* isomers. In the absence of standards for *cis* isomers, these compounds were first characterized based on their λ*_cis_* in addition to their relative retention times given in previous studies [20,22,26]. In general, the combined data from LC, UV-vis, MS and NMR enable us to identify and confirm the identity of free lutein and lutein mono- and di-ester isomers and other carotenoids.

Other carotenoid compounds found in HCS were lutein 3′-*O*-oleate, lutein 3-*O*-oleate, lutein 3′-*O*-palmitate, lutein 3-*O*-palmitate and lutein di-linoleate (Table 1). Once again the fragmentation pattern and UV-vis data were used in the identification of those compounds. The most abundant in-source fragment for lutein 3′-*O*-oleate was ion at *m*/*z* 551, corresponding to the loss of oleic acid on the ε-ionone ring, compared with [M-H_2_O+H]^+^ ion at *m*/*z* 816 for lutein 3*-O-*oleate. There were also similar differences in the mass spectra between lutein 3′-*O*-palmitate and lutein 3-*O*-palmitate (Appendix A). For lutein 3′-*O*-palmitate, the most abundant fragment was [M-FA+H]^+^ ion at *m*/*z* 551, while lutein 3-*O*-palmitate had the [M-H_2_O+H]^+^ ion at *m*/*z* 790 as the major fragment. The preferential loss of fatty acid from lutein 3′-*O*-palmitate or the loss of water from lutein 3-*O*-palmitate is from the acylated 3′-hydroxyl in the ε-ionone ring as previously reported [20]. Lutein di-linoleate appeared as the molecular ion at *m*/*z* 1093 and the fragment [M-2FA+H]^+^ at *m*/*z* 533 under APCI^+^ ionization mode, with the molecular fragment being the prevailing ion (Figure 3C). The loss of a fatty acid from each ε-ionone and β-ionone ring produced the fragment *m*/*z* 533, which was found at a lower intensity; we did not observe an expected [M-FA+H]^+^ fragment, potentially due to interfering non-carotenoid peaks (Figure 3) [25]. These peaks could arise from tiny concentrations of oil impurities in the extracts. Other studies have shown the presence of triacylglycerols in seed extracts, which ionize extremely well under the same positive APCI and ESI conditions as carotenoids and can interfere with their detection [27,28]. Additionally, HCS is rich in oil that contains high amounts of oleic, linoleic and palmitic fatty acids [29]. It will be of interest if more investigations using different analytical techniques are performed to confirm the identity of fatty acids and *cis* isomers.

### 3.2. Quality of Muffins and Breads

In the current study, prototype muffin and bread products made from HCS alone or in blends with corn and wheat were developed to determine their acceptability and the effect of the baking process on carotenoid composition. The appearance of muffins and breads is shown in Appendix A. In order to ensure acceptability of the developed baked products and their potential market, it was necessary to evaluate the quality of the products. Three low-fat muffins (HCS, HCS/corn 1:1, and 1:2) were assessed based on their external and internal quality and nutrient content (Appendix A). Corn was incorporated into the muffin formulas due to its high concentration of carotenoids, especially lutein and zeaxanthin [22,30]. The three muffins were rated acceptable based on their height, and internal and external quality. The HCS muffin was considered a good source of protein (17.9%) and total dietary fiber (12.8%). This will provide a protein-rich functional food for the gluten-free food market.

Three bread products baked from composite flours of wheat and HCS were compared with a bread control (100% wheat) based on loaf volume, sensory properties and nutrient content (Appendix A). Replacement of wheat flour with HSC flour up to 25% by weight was acceptable and comparable with the control bread. These results are in agreement with previous research [16]. Breads made from wheat and HCS composite flour (50/50, *w*/*w*) had substantially lower loaf volume and sensory properties compared with the control and other breads. The HCS containing breads had higher protein and total dietary fiber contents compared with the control bread due to the addition of HCS wholegrain flour. These bread products could be a good source of protein, dietary fiber, and bioactive compounds.

### 3.3. Changes in Carotenoid Composition in Muffins

The role of lutein in the health of the eye and brain has been shown previously. Nonetheless, studies have shown a dietary gap between the daily intake (1.0–1.8 mg/day) [31] and the suggested effective dose of lutein estimated based on reducing the risk of cataracts and AMD (6 mg/day) [1]. This indicates a need for making high lutein functional foods in addition to commercially available lutein supplements, especially since no daily intake for lutein is recommended. Muffins made from HCS alone or in blends with corn could help in filling this dietary gap. Table 2 shows the main carotenoids in HCS and its blends with corn and the impact of the baking process on carotenoids. All-*trans* lutein, along with its *cis* isomers and mono- and di-esters was found to constitute the major portion of carotenoids in HCS. Altogether they made up to about 83% of the total carotenoids in HCS. All-*trans* lutein was the principal carotenoid (3.1 µg/g) followed by lutein 3-*O*-linoleate (1.8 µg/g), lutein 3-*O*-oleate (0.6 µg/g) and lutein di-linoleate (0.6 µg/g). Free or unbound lutein and lutein esters are absorbed from foods and dietary supplements, but the ester form requires prior de-esterification by intestinal enzymes [32]. A study on HCS genotypes has found β-carotene as the main carotenoid followed by lutein and zeaxanthin [33]. 

We detected trivial amount of β-carotene in HCS, which is in disagreement with the latter study. It is possible that β-carotene could co-elute with other carotenoids in the grain, which could lead to misidentification. In the current study, the identity of carotenoid compounds was identified and confirmed based on authentic standards, including β-carotene and lutein esters and the use of several structural properties. The corn flour used in the current study contained higher amounts of lutein (4.5 µg/g) and zeaxanthin (5.9 µg/g) compared with HCS. Since lutein and its cousin zeaxanthin constitute the macular pigments, foods that are rich in lutein, zeaxanthin, and their derivatives would boost their daily intake. Blending HCS with corn at ratios of 1:1 and 1:2 increased total unbound carotenoids by about 105% and 133%, respectively, while it decreased total bound carotenoids by 53% and 59%, respectively. These changes are due to the significant increase in the unbound or free carotenoids (e.g., lutein from 3.1 to 4.1 and 4.5 µg/g & zeaxanthin from 0.4 to 3.6 and 4.2 µg/g at ratios 1:1 and 1:2, respectively) due to the addition of cornflour. The addition of corn also resulted in an increase in total carotenoids from 7.5 µg/g in HCS to 10.3–11.3 µg/g in the blends. Currently, there are corn varieties with very high levels of lutein and/or zeaxanthin which could be used to boost lutein in food products even more.

The muffin-making process resulted in significant reductions in all carotenoid compounds but at different extents subject to carotenoid type (Table 2). For example, all *trans*-lutein had the highest reduction among carotenoids. The batter preparation resulted in higher reductions in unbound, bound, and total carotenoids than that of oven baking of muffins, e.g., 26 vs. 19%, 38 vs. 16%, and 31 vs. 17% for HCS muffin product, respectively. Similar trends were also observed for muffins made from HCS and corn blends. The total carotenoid loss had a range of 48–53% for the three muffin products. Interestingly, the reduction percent of total unbound carotenoids was much lower than that of bound carotenoids in HCS, while an opposite trend was observed in muffins made from HCS and corn blends. During the preparation of the batter, carotenoids could undergo changes due to enzyme degradation and oxidation, while during oven baking thermal degradation and isomerization could take place. Previous research has reported reductions in lutein and zeaxanthin in muffins made from einkorn wheat by about 64 and 57%, respectively, due to dough preparation and oven baking [34].

### 3.4. Changes in Carotenoid Composition in Breads

Bread is one of the world’s most important staple food and is often consumed every day. Thus the availability of nutritious breads could make a difference in human health through healthy eating choices. The current study used HCS as a source of bioactive peptides, minerals, vitamins, carotenoids, and polyphenols to replace part of wheat flour in the bread-making process. The addition of HCS increased total carotenoids from 1.5 µg/g in wheat flour to 2.7, 3.8 and 4.9 µg/g at replacement levels of 15, 25 and 50%, respectively (Table 3). There were significant changes in carotenoids during dough formation and oven baking. Total carotenoids decreased by about 67, 48, 50 and 47% in control, 15, 25 and 50% breads, respectively. This finding indicates that degradation of carotenoids is dependent on carotenoid type; the higher amount of bound carotenoids, the more stable they remain during the baking process. The control bread, which had no bound carotenoids, exhibited the highest reduction percentage. Esterification of carotenoid with fatty acids in bound carotenoids affect their stability and behavior during processing and storage [35]. Similar to muffin, dough formation resulted in higher reductions in unbound, bound and total carotenoids compared with oven baking of breads in all bread products. It has been reported that dough preparation causes a more pronounced loss of carotenoids and tocols (51.5 and 33.0%) compared to baking (22.5 and 9.1%) in flatbread [36]. The study has also reported that carotenoids are less stable than tocols, and the presence of tocols could improve stability of carotenoids during the baking process. Carotenoids undergo several biochemical and chemical reactions during dough preparation including enzyme degradation and oxidation which contribute to their loss. During oven baking of breads, additional but smaller reductions in total carotenoids occurred, e.g., 20 vs. 47%, 11 vs. 37%, 8 vs. 42% and 8 vs. 39% for breads made form wheat, wheat/HCS (85/15), wheat/HCS (75/25) and wheat/HCS (50/50), respectively (Table 3). Carotenoids are sensitive to light and temperature through oxidation and isomerization, resulting in a partial loss of carotenoids. The replacement of wheat flour with HCS up to 25% gave acceptable breads which would increase protein, carotenoids and other bioactive compounds in the end product.

## 4. Conclusions

Hairless canary seed is a true cereal grain with unique nutritional profile that makes it a suitable functional food ingredient. For the first time detailed composition of carotenoids in HCS has been identified based on UPLC and LC-MS/MS analyses. Hairless canary seed and corn were fairly rich in carotenoids compared with other cereal grains having a total content of 7.6 and 12.9 µg/g, respectively. HCS is rich in lutein and its mono- and di-esters especially lutein 3-*O*-linoleate, lutein 3-*O*-oleate and lutein di-linoleate, while corn is rich in free lutein and zeaxanthin. Thus, they might complement each other in their carotenoid composition in terms of lutein stability and bioavailability. Baking process resulted in significant reductions in carotenoids in muffin and bread products. It appears that batter or dough preparation causes more reductions in carotenoids than oven baking probably due to enzymatic oxidation and degradation. In addition, muffin-making resulted in lower lutein reductions compared to the bread-making process. Thus, muffins made from hairless canary seed alone or in blends with corn could boost the daily intake of lutein and/or zeaxanthin. Further research is underway to assess the bioavailability of carotenoids in HCS muffin and bread products.

## Figures and Tables

**Figure 1 molecules-27-01307-f001:**
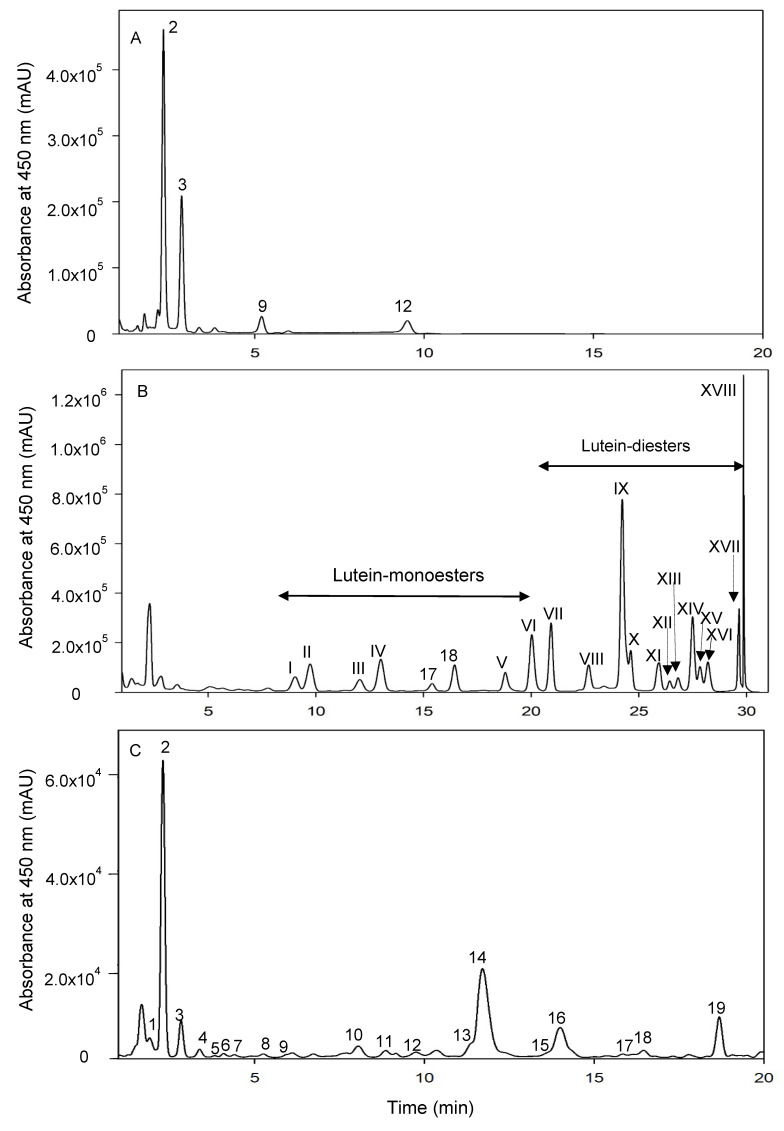
UPLC chromatograms of authentic carotenoid standards (**A**), lab-prepared mono- and di-esters of lutein standards (**B**) and canary seed extract (**C**). Numbers 1–19 correspond to compounds in Table 1, and compounds I-XVIII are I (3′-*O*-laurate), II (3-*O*-laurate), III (3′-*O*-myristate), IV (3-*O*-myristate), V (3′-*O*-stearate), VI (3-*O*-stearate), VII (dilaurate), VIII (co-eluting 3-O-myristate-3′-O-laurate and 3-O-laurate-3′-O-myristate), IX (co-eluting dimyristate and 3-O-myristate-3′-O-laurate), X (3-O-laurate-3′-O-myristate), XI (co-eluting 3-O-myristate-3′-O-palmitate and 3-O-palmitate-3′-O-myristate), XII (3-O-laurate-3′-O-stearate), XIII (3-O-stearate-3′-O-laurate), XIV (dipalmitate), XV (3-O-myristate-3′-O-stearate), XVI (3-O-stearate-3′-O-myristate), XVII (co-eluting 3-O-palmitate-3′-O-stearate and 3-O-stearate-3′-O-palmitate), XVIII (distearate).

**Figure 2 molecules-27-01307-f002:**
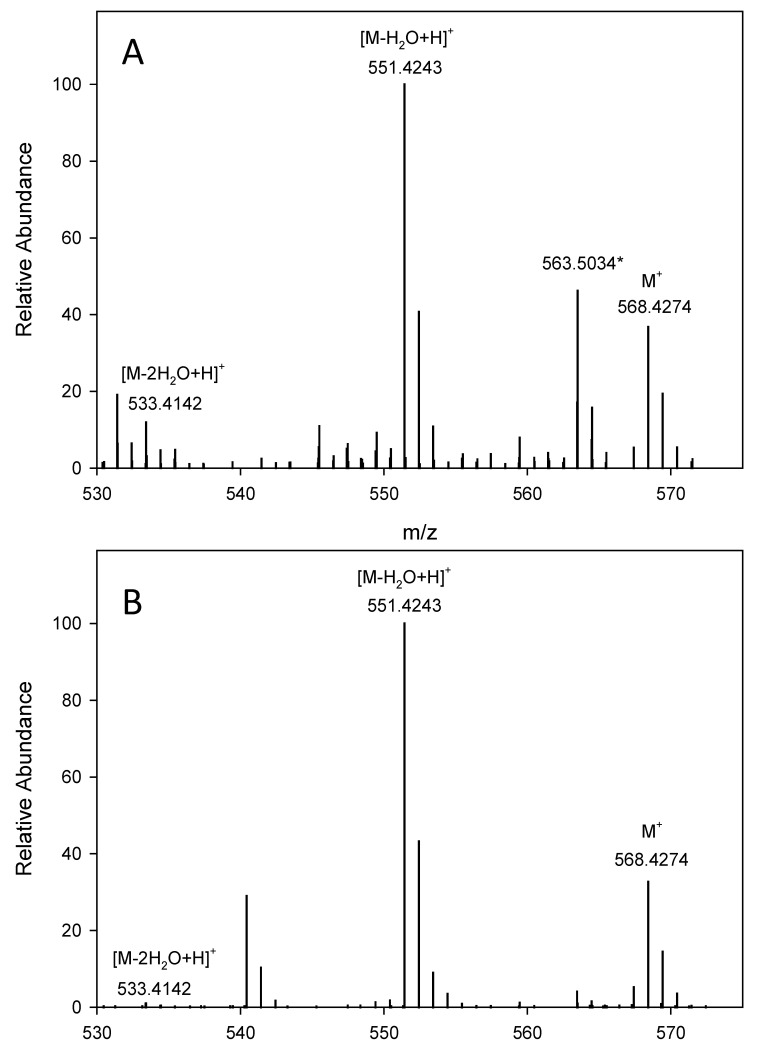
MS spectra of all-*trans*-lutein in two positive ionization modes, (**A**) APCI and (**B**) ESI. * Major non-carotenoid peaks.

**Figure 3 molecules-27-01307-f003:**
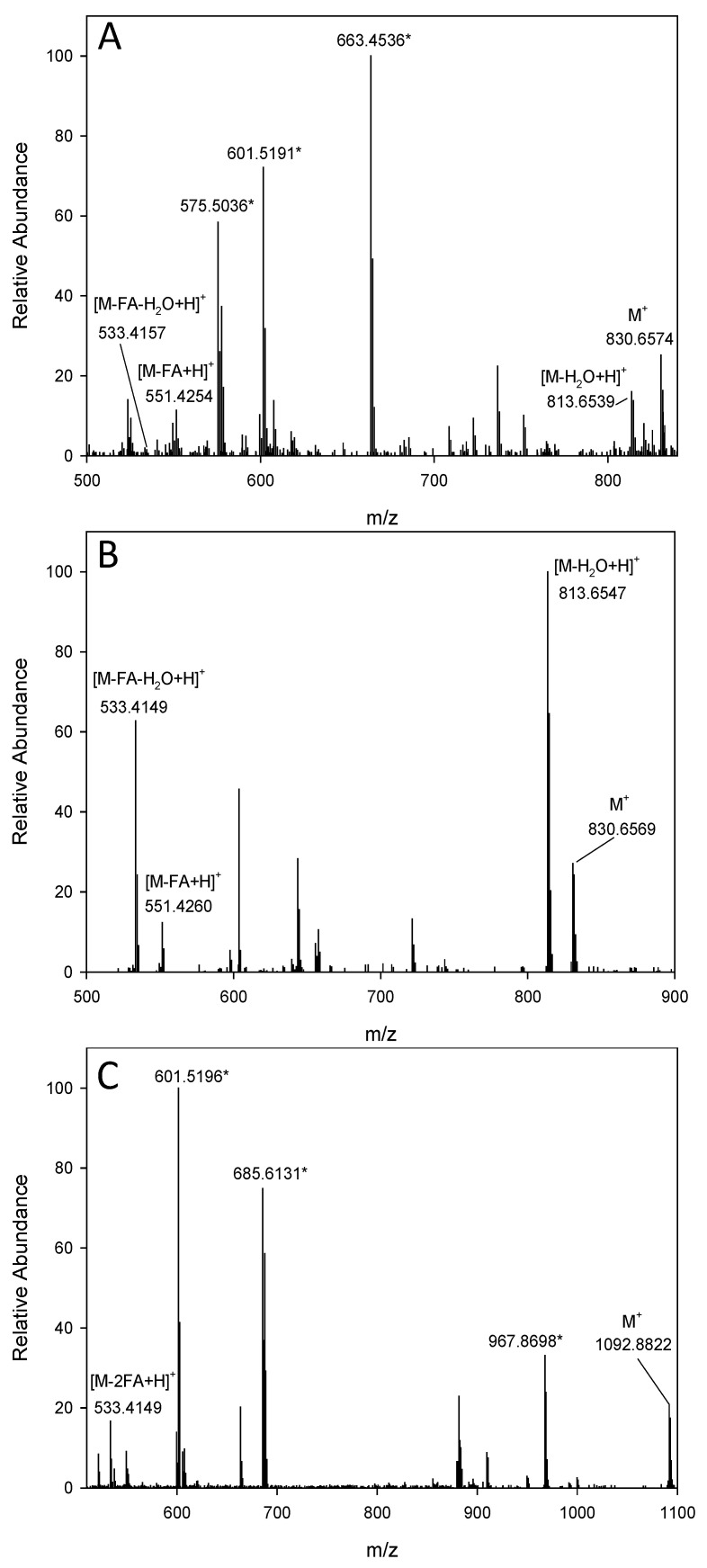
MS spectra of trans-lutein mono- and di-esters identified in canary seed samples in positive ionization modes, (**A**) trans-lutein-3′-O-linoleate (ESI), (**B**) trans-lutein-3-O-linoleate (APCI), (**C**) trans-lutein-dilinoleate (APCI). * Major non-carotenoid peaks.

**Table 1 molecules-27-01307-t001:** Detailed composition of carotenoids identified in hairless canary seed and their chromatographic and spectrometric properties.

Peak Number #	Retention Time (min)	Identity of Compound	Formula	Accurate Mass (g/mol)	MS in-Source Fragmentation, *m*/*z* (Relative Abundance, %) ^c^	UV-Vis Absorption Bands, λ_I_, λ_II_ (λ_max_) and λ_III_ (nm) ^d^
1	1.9	15-*cis*-Lutein	C_40_H_56_O_2_	568.43	551.43 (100), 568.43 (40)	416, 436, 464
2	2.3	all-*trans*-Lutein ^a^	C_40_H_56_O_2_	568.43	551.42 (100), 568.43 (38), 533.42 (13)	418, 444, 472
3	2.8	all-*trans*-Zeaxanthin ^a^	C_40_H_56_O_2_	568.43	569.44 (100), 551.42 (44)	424, 446, 474
4	3.4	9-*cis*-Lutein	C_40_H_56_O_2_	568.43	551.43 (100), 568.43 (15)	415, 440, 464
5	3.8	9-*cis*-Zeaxanthin	C_40_H_56_O_2_	568.43	569.43 (100), 551.42 (86)	420, 444, 472
6	4.1	15-*cis-*β-Cryptoxanthin	C_40_H_56_O	552.43	553.44 (100), 535.43 (14)	420, 444, 472
7	4.4	13-*cis-*β-Cryptoxanthin	C_40_H_56_O	552.43	553.44 (100), 535.43 (37)	424, 448, 476
8	5.0	13′-*cis-*β-Cryptoxanthin.	C_40_H_56_O	552.43	553.44 (100), 535.43 (37)	416, 442, 466
9	5.8	all-*trans*-β-Cryptoxanthin ^a^	C_40_H_56_O	552.43	553.44 (100), 535.43 (15)	423, 450, 474
10	8.1	15-*cis-*Lutein-3-*O*-linoleate	C_58_H_86_O_3_	830.66	813.65 (100), 533.41 (67), 830.66 (28), 551.41 (16)	414, 436, 464
11	8.9	13-*cis-*Lutein-3-*O*-linoleate	C_58_H_86_O_3_	830.66	813.66 (100), 533.41 (64), 830.66 (28), 551.43 (14)	412, 436, 464
12	10.7	all-*trans*-β-Carotene ^a^	C_40_H_56_	536.44	537.44 (100)	423, 450, 474
13	11.4	Lutein 3′-*O*-linoleate ^b^	C_58_H_86_O_3_	830.66	830.66 (100), 551.43 (32), 813.65 (2)	417, 444, 472
14	11.7	Lutein-3-*O*-linoleate	C_58_H_86_O_3_	830.66	813.66 (100), 533.42 (63), 830.66 (24), 551.43 (12)	417, 444, 472
15	13.7	Lutein-3′-*O*-oleate ^b^	C_58_H_88_O_3_	832.67	832.67 (100), 551.42 (43)	419, 444, 472
16	14.0	Lutein-3-*O*-oleate ^b^	C_58_H_88_O_3_	832.67	832.67 (100), 815.67 (53), 533.42 (7)	419, 444, 472
17	15.9	Lutein-3′-*O*-palmitate ^a^	C_56_H_86_O_3_	806.66	551.43 (100), 806.66 (36), 533.41 (17), 789.65 (4)	420, 444, 472
18	16.5	Lutein 3-*O*-palmitate ^a^	C_56_H_86_O_3_	806.66	789.65 (100), 533.42 (78), 806.66 (56), 551.43 (19)	420, 444, 472
19	18.7	Lutein dilinoleate	C_76_H_116_O_4_	1092.89	1092.88 (100), 533.41 (77)	419, 444, 472

^a^ Carotenoids that are identified using authentic carotenoid standards. ^b^ Carotenoids that are identified using ESI positive mode. ^c^ Relative abundance is calculated with respect to the major in-source fragment of identified carotenoid using either APCI or ESI positive mode. ^d^ *cis*-Carotenoids exhibit additional peaks (λ_cis_) around 330–340 nm. Peak numbers match those in Figure 1.

**Table 2 molecules-27-01307-t002:** Content of major carotenoids in muffin formulations and products (µg/g) and the impact of baking process on carotenoids (% decrease) ^x^.

Carotenoids	Hairless Canary Seed (100%)			Hairless Canary Seed/Corn Blend (1:1, *w*/*w*)			Hairless Canary Seed/Corn Blend (1:2, *w*/*w*)		
	Dry Flour	Batter	Muffin	Dry Flour	Batter	Muffin	Dry Flour	Batter	Muffin
15-*cis*-Lutein	0.06 ± 0.02 ^a^	0.04 ± 0.01 ^a^	0.05 ± 0.01 ^a^	0.03 ± 0.01 ^a^	0.03 ± 0.01 ^a^	0.07 ± 0.01 ^a^	0.1 ± 0.03 ^a^	0.1 ± 0.01 ^a^	0.1 ± 0.01 ^a^
all-*trans*-Lutein	3.1 ± 0.17 ^a^	2.3 ± 0.15 ^b^	1.5 ± 0.06 ^c^	4.1 ± 0.04 ^a^	2.5 ± 0.10 ^b^	1.7 ± 0.04 ^c^	4.5 ± 0.08 ^a^	2.8 ± 0.03 ^b^	2.0 ± 0.05 ^c^
all-*trans*-Zeaxanthin	0.4 ± 0.02 ^a^	0.3 ± 0.01 ^b^	0.2 ± 0.01 ^c^	3.6 ± 0.08 ^a^	2.0 ± 0.05 ^b^	1.5 ± 0.03 ^c^	4.2 ± 0.02 ^a^	2.6 ± 0.08 ^b^	2.2 ± 0.05 ^c^
9-*cis*-Lutein	0.1 ± 0.01 ^a^	0.2 ± 0.01 ^a^	0.2 ± 0.02 ^a^	0.1 ± 0.01 ^a^	0.1 ± 0.01 ^a^	0.1 ± 0.01 ^a^	0.2 ± 0.01 ^a^	0.2 ± 0.01 ^a^	0.1 ± 0.01 ^b^
9-*cis*-Zeaxanthin	0.1 ± 0.01 ^a^	0.1 ± 0.01 ^a^	0.1 ± 0.01 ^a^	0.2 ± 0.01 ^a^	0.1 ± 0.01 ^a^	0.1 ± 0.01 ^a^	0.3 ± 0.01 ^a^	0.2 ± 0.01 ^b^	0.2 ± 0.01 ^b^
15-*cis-*β-Cryptoxanthin	0.3 ± 0.02 ^a^	0.2 ± 0.01 ^b^	0.2 ± 0.01 ^b^	0.4 ± 0.01 ^a^	0.2 ± 0.01 ^b^	0.2 ± 0.01 ^b^	0.3 ± 0.01 ^a^	0.2 ± 0.01 ^b^	0.2 ± 0.01 ^b^
all-*trans*-β-Cryptoxanthin	0.2 ± 0.02 ^a^	0.1 ± 0.01 ^b^	0.1 ± 0.01 ^b^	0.3 ± 0.02 ^a^	0.2 ± 0.02 ^b^	0.2 ± 0.02 ^b^	0.4 ± 0.01 ^a^	0.2 ± 0.01 ^b^	0.2 ± 0.01 ^b^
Lutein-3-*O*-linoleate	1.8 ± 0.06 ^a^	1.3 ± 0.07 ^b^	1.0 ± 0.07 ^c^	0.8 ± 0.01 ^a^	0.6 ± 0.01 ^b^	0.4 ± 0.01 ^c^	0.8 ± 0.06 ^a^	0.6 ± 0.03 ^b^	0.5 ± 0.04 ^c^
Lutein-3-*O*-oleate	0.6 ± 0.03 ^a^	0.4 ± 0.02 ^b^	0.3 ± 0.01 ^b^	0.5 ± 0.05 ^a^	0.4 ± 0.03 ^b^	0.4 ± 0.03 ^b^	0.3 ± 0.03 ^a^	0.2 ± 0.01 ^b^	0.2 ± 0.02 ^b^
Lutein dilinoleate	0.6 ± 0.02 ^a^	0.3 ± 0.02 ^b^	0.2 ± 0.01 ^b^	0.2 ± 0.02 ^a^	0.1 ± 0.01 ^a^	0.1 ± 0.01 ^a^	0.2 ± 0.02	0.1 ± 0.01 ^a^	0.1 ± 0.01 ^a^
Total unbounds (free)	4.3	3.2	2.4	8.8	5.1	3.9	10.0	6.3	5.0
Total bounds (mono- and di-esters)	3.2	2.0	1.5	1.5	1.1	0.9	1.3	0.9	0.8
Total carotenoids	7.5	5.2	3.9	10.3	6.2	4.8	11.3	7.2	5.8
% Decrease									
Total unbounds	-	25.6	44.2 (18.6) ^y^	-	42.0	55.7 (13.7)	-	37.0	50.0 (13.0)
Total bounds	-	37.5	53.1 (15.6)	-	26.7	40.0 (13.3)	-	30.8	38.5 (7.7)
Total carotenoids	-	30.7	48.0 (17.3)	-	39.8	53.4 (13.6)	-	36.3	48.7 (12.4)

^x^ For each product and compound, mean values in a row followed by a different superscript letter are significantly different at *p* < 0.05. ^y^ Figures between brackets are reduction percent due to oven baking.

**Table 3 molecules-27-01307-t003:** Content of major carotenoids in bread formulations and products (µg/g) and the impact of baking process on carotenoids (% decrease) ^x^.

Carotenoids	Wheat (100%)			Wheat/Hairless Canary Seed (85/15, *w*/*w*)			Wheat/Hairless Canary Seed (75/25, *w*/*w*)			Wheat/Hairless Canary Seed (50/50, *w*/*w*)		
	Dry Flour	Dough	Bread	Dry Flour	Dough	Bread	Dry Flour	Dough	Bread	Dry Flour	Dough	Bread
15-*cis*-Lutein	0.1 ± 0.01 ^a^	0.05 ± 0.01 ^a^	0.04 ± 0.01 ^a^	0.05 ± 0.01 ^a^	0.03 ± 0.01 ^a^	0.03 ± 0.01 ^a^	0.05 ± 0.01 ^a^	0.03 ± 0.01 ^a^	0.03 ± 0.01 ^a^	0.1 ± 0.02 ^a^	0.06 ± 0.02 ^a^	0.06 ± 0.02 ^a^
all-*trans*-Lutein	1.0 ± 0.02 ^a^	0.5 ± 0.03 ^b^	0.3 ± 0.01 ^c^	1.2 ± 0.04 ^a^	0.6 ± 0.03 ^b^	0.5 ± 0.03 ^c^	1.7 ± 0.05 ^a^	0.8 ± 0.02 ^b^	0.7 ± 0.01 ^c^	2.2 ± 0.07 ^a^	0.9 ± 0.07 ^b^	0.8 ± 0.03 ^c^
all-*trans*-Zeaxanthin	0.4 ± 0.02 ^a^	0.2 ± 0.03 ^b^	0.2 ± 0.02 ^b^	0.4 ± 0.02 ^a^	0.3 ± 0.03 ^b^	0.3 ± 0.02 ^b^	0.4 ± 0.02 ^a^	0.2 ± 0.01 ^b^	0.2 ± 0.02 ^b^	0.4 ± 0.04 ^a^	0.3 ± 0.01 ^b^	0.3 ± 0.01 ^b^
Lutein-3-*O*-linoleate	nd	nd	nd	0.5 ± 0.02 ^a^	0.3 ± 0.03 ^b^	0.3 ± 0.03 ^b^	0.6 ± 0.02 ^a^	0.4 ± 0.03 ^b^	0.4 ± 0.03 ^b^	1.1 ± 0.06 ^a^	0.7 ± 0.01 ^b^	0.6 ± 0.03 ^c^
Lutein-3-*O*-oleate	nd	nd	nd	0.4 ± 0.02 ^a^	0.3 ± 0.03 ^b^	0.2 ± 0.03 ^c^	0.5 ± 0.02 ^a^	0.4 ± 0.03 ^b^	0.3 ± 0.03 ^c^	0.7 ± 0.06 ^a^	0.5 ± 0.03 ^b^	0.4 ± 0.03 ^c^
Lutein dilinoleate	nd	nd	nd	0.3 ± 0.02 ^a^	0.2 ± 0.01 ^b^	0.1 ± 0.01 ^c^	0.5 ± 0.02 ^a^	0.4 ± 0.02 ^b^	0.3 ± 0.02 ^c^	0.6 ± 0.05 ^a^	0.5 ± 0.02 ^b^	0.4 ± 0.02 ^c^
Total unbounds (free)	1.5	0.8	0.5	1.5	0.9	0.8	2.2	1.0	0.9	2.5	1.3	1.2
Total bounds (mono- and di-esters)	0.0	0.0	0.0	1.2	0.8	0.6	1.6	1.2	1.0	2.4	1.7	1.4
Total carotenoids	1.5	0.8	0.5	2.7	1.7	1.4	3.8	2.2	1.9	4.9	3.0	2.6
% Decrease												
Total unbounds	-	46.7	66.7 (20.0) ^y^	-	40.0	46.7 (6.7)	-	54.5	59.1 (4.6)	-	48.0	52.0 (4.0)
Total bounds	-	-	-	-	33.3	50.0 (16.7)	-	25.0	37.5 (12.5)	-	29.2	41.7 (12.5)
Total carotenoids	-	46.7	66.7 (20.0)	-	37.0	48.1 (11.1)	-	42.1	50.0 (7.9)	-	38.8	46.9 (8.1)

^x^ For each product and carotenoid mean values in a row followed by a different superscript letter are significantly different at *p* < 0.05. y Figures between brackets are reduction percent due to oven baking.

## Data Availability

Not applicable.

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
