# Peer review of "Identification of Carotenoids in Hairless Canary Seed and the Effect of Baking on Their Composition in Bread and Muffin Products"

_molecules, 2022, doi:10.3390/molecules27041307_

Round 1

Reviewer 1 Report

After reviewing the manuscript entitled Identification of carotenoids in hairless canary seed and the effect of baking on their composition in bread and muffin products, the following recommendations are made.

Line 1: insert the word article one line before the title, as indicated in the Microsoft word template (MWT).

Line 10: use the format style according to the MWT (* Correspondence: [email protected]; Tel.: ...).

Line 11: remove bold text formatting for the word carotenoids.

Line 32: use the hyphen (–) instead of the hyphen (-), to indicate a number range in references, as indicated in MWT. e.g., [4–6]. Modify throughout the manuscript.

Line 90, 100: insert space (400 °F).

Line 93: delete space after parenthesis… Canada) .

Line 90: insert space (950 °F).

Line 110: use the font size format in number 10 for the equation.

Line 120: insert space (0.5 g).

Line 125: insert space (4.6 mm).

Line 128: insert space (35 °C). Is it allowed to use two different temperature units in the text? e.g., °F or °C.

Line 140: insert space between % and MeOH, FA and MtBE.

Line 141-143: remove space in time ranges as in lines 127 and 128. e.g., 0-10 min

Line 144: insert space (35 °C).

Line 153-154: in figure 1, is the font in the graphic Palatino linotype?

Lines 215 and 216: is correct the symbol ß ?.

Line 228-229: in figure 2, is the font in the graphic Palatino linotype?

Line 233:  sometimes appears as APCI+ and ESI+ or APCI+ and ESI+, it is necessary to standardize the text format in the document.

Line 423: according to the MWT, it is necessary to include the digital object identifier (DOI) for all references where available. Also, the titles of each reference must be written in lowercase text format, except for the first letter of the first word. A space must be inserted between the reference number and the last name of the first author. Review in detail the indications described in MWT.

In the statistical analysis, I consider that it would be better to use a two-way factorial, in which the treatment and each stage of the process should be considered as fixed factors (e.g., dry flour, batter, and moffin; or dry flour, dough, and bread), which would reduce the error in the design. Once the analysis is done, you could infer the results based on the single and double interactions.

Otherwise, a multivariate analysis (eg, principal component analysis) could be used to graphically observe how each of the compounds identified in the products are grouped due to the effect of the processing stage (e.g., dry flour, batter, and muffin; or dry flour, dough, and bread).

Author Response

Responses to reviewer’s comments

Reviewer 1

Line 1: insert the word article one line before the title, as indicated in the Microsoft word template (MWT).

Changes made

Line 10: use the format style according to the MWT (* Correspondence: [email protected]; Tel.: ...).

Changes made

Line 11: remove bold text formatting for the word carotenoids.

Bold removed

Line 32: use the hyphen (–) instead of the hyphen (-), to indicate a number range in references, as indicated in MWT. e.g., [4–6]. Modify throughout the manuscript.

Changes made

Line 90, 100: insert space (400 °F).

Changes made

Line 93: delete space after parenthesis… Canada).

Deleted

Line 90: insert space (950 °F).

Changes made

Line 110: use the font size format in number 10 for the equation.

Done

Line 120: insert space (0.5 g).

Done

Line 125: insert space (4.6 mm).

Done

Line 128: insert space (35 °C). Is it allowed to use two different temperature units in the text? e.g., °F or °C.

Done

Line 140: insert space between % and MeOH, FA and MtBE.

Done

Line 141-143: remove space in time ranges as in lines 127 and 128. e.g., 0-10 min

Done

Line 144: insert space (35 °C).

Done

Line 153-154: in figure 1, is the font in the graphic Palatino linotype?

Yes

Lines 215 and 216: is correct the symbol ß ?.

Changed into the correct symbol.

Line 228-229: in figure 2, is the font in the graphic Palatino linotype?

Yes

Line 233:  sometimes appears as APCI+ and ESI+ or APCI+ and ESI+, it is necessary to standardize the text format in the document.

Done

Line 423: according to the MWT, it is necessary to include the digital object identifier (DOI) for all references where available. Also, the titles of each reference must be written in lowercase text format, except for the first letter of the first word. A space must be inserted between the reference number and the last name of the first author. Review in detail the indications described in MWT.

The reference format is applied as per the journal instructions and our last publication in 2021.

In the statistical analysis, I consider that it would be better to use a two-way factorial, in which the treatment and each stage of the process should be considered as fixed factors (e.g., dry flour, batter, and moffin; or dry flour, dough, and bread), which would reduce the error in the design. Once the analysis is done, you could infer the results based on the single and double interactions.

Otherwise, a multivariate analysis (eg, principal component analysis) could be used to graphically observe how each of the compounds identified in the products are grouped due to the effect of the processing stage (e.g., dry flour, batter, and muffin; or dry flour, dough, and bread).

One way ANOVA is used to achieve the study goal of looking into behavior of carotenoids during baking process (differences between technological treatments) and to fulfil the experimental design (CRD). The objective was not to determine differences among different formulations or baked products but to develop high carotenoid foods.

Reviewer 2 Report

Lutein is an antioxidant that has gathered increasing attention due to its potential role in preventing or ameliorating age-related macular degeneration. Currently, it is produced from marigold oleoresin, but continuous reports of lutein-producing microalgae pose the question if those microorganisms can become an alternative source. Authors emphasized hairless canary seed (HCS) as novel food for theses kind of pigments that can be used in bakery products.  

I have few comments to improve this article:

Please clarify this statement, as it appears that this pigment is harmful, causing yellow spots in the retina. At the center of the retina, there is a yellow oval area (back of the eye). It's the region of the retina that gives you clear, detailed vision in the center. “These pigments constitute the yellow spot in the human retina and are referred to as macular pigments”.

Please draw a comparative table with other sources of lutein that can be used in bakery products.

Reviewer 3 Report

This research is necessary in order to develop canary seed as a functional food therefore it should be published. The manuscript requires some further editing including:

Many spacing errors and some capitalization errors

line 90, 98-100 temps should be in C not F

Line 233 no . after Fig 3

line 307 from not form

line 300 repeats information discussed in Introduction

lines 324, 364 others muffin and food are used; they should be plural

lines 359-360 incorrect grammar--does not make sense; rewrite

362 reductions from what process?  Baking, batter prep etc.  Please clarify

Table 3 not formatted correctly

In addition, some discussion of the carotenoids that could have been added by the corn oil should be provided.  If it is insignificant, fine just mention that. I would also like a few lines regarding the agronomic yields and production costs for this crop. If the crop is very costly, 25% addition is interesting but not feasible from a commercial standpoint.

Round 2

Reviewer 1 Report

The requested changes were met, however:
1. Line 517, this reference includes the DOI. However, in the other references it does not appear.
2. The titles of the references must appear in lowercase letters.